# Immune-Mediated Ocular Surface Disease in Diabetes Mellitus—Clinical Perspectives and Treatment: A Narrative Review

**DOI:** 10.3390/biomedicines12061303

**Published:** 2024-06-12

**Authors:** Laura Andreea Ghenciu, Ovidiu Alin Hațegan, Sorin Lucian Bolintineanu, Alexandra-Ioana Dănilă, Alexandra Corina Faur, Cătălin Prodan-Bărbulescu, Emil Robert Stoicescu, Roxana Iacob, Alina Maria Șișu

**Affiliations:** 1Department of Functional Sciences, “Victor Babes” University of Medicine and Pharmacy Timisoara, Eftimie Murgu Square No. 2, 300041 Timisoara, Romania; bolintineanu.laura@umft.ro; 2Discipline of Anatomy and Embriology, Medicine Faculty, ‘Vasile Goldis’ Western University of Arad, Revolution Boulevard 94, 310025 Arad, Romania; 3Department of Anatomy and Embriology, “Victor Babes” University of Medicine and Pharmacy Timisoara, Eftimie Murgu Square No. 2, 300041 Timisoara, Romania; s.bolintineanu@umft.ro (S.L.B.); alexandra.danila@umft.ro (A.-I.D.); faur.alexandra@umft.ro (A.C.F.); catalin.prodan-barbulescu@umft.ro (C.P.-B.); roxana.iacob@umft.ro (R.I.); alinasisu@umft.ro (A.M.Ș.); 4Doctoral School, “Victor Babes” University of Medicine and Pharmacy Timisoara, Eftimie Murgu Square No. 2, 300041 Timisoara, Romania; 5IInd Surgery Clinic, “Victor Babes” University of Medicine and Pharmacy Timisoara, Eftimie Murgu Square No. 2, 300041 Timisoara, Romania; 6Field of Applied Engineering Sciences, Specialization Statistical Methods and Techniques in Health and Clinical Research, Faculty of Mechanics, ‘Politehnica’ University Timisoara, Mihai Viteazul Boulevard No. 1, 300222 Timisoara, Romania; stoicescu.emil@umft.ro; 7Department of Radiology and Medical Imaging, “Victor Babes” University of Medicine and Pharmacy Timisoara, Eftimie Murgu Square No. 2, 300041 Timisoara, Romania; 8Research Center for Pharmaco-Toxicological Evaluations, “Victor Babes” University of Medicine and Pharmacy Timisoara, Eftimie Murgu Square No. 2, 300041 Timisoara, Romania

**Keywords:** ocular surface disease, diabetic neuropathy, ocular immunology, diabetic keratopathy, limbal stem cell therapy, gene therapy

## Abstract

Diabetes mellitus (DM) is a chronic metabolic disorder marked by hyperglycemia due to defects in insulin secretion, action, or both, with a global prevalence that has tripled in recent decades. This condition poses significant public health challenges, affecting individuals, healthcare systems, and economies worldwide. Among its numerous complications, ocular surface disease (OSD) is a significant concern, yet understanding its pathophysiology, diagnosis, and management remains challenging. This review aims to explore the epidemiology, pathophysiology, clinical manifestations, diagnostic approaches, and management strategies of diabetes-related OSD. The ocular surface, including the cornea, conjunctiva, and associated structures, is vital for maintaining eye health, with the lacrimal functional unit (LFU) playing a crucial role in tear film regulation. In DM, changes in glycosaminoglycan metabolism, collagen synthesis, oxygen consumption, and LFU dysfunction contribute to ocular complications. Persistent hyperglycemia leads to the expression of cytokines, chemokines, and cell adhesion molecules, resulting in neuropathy, tear film abnormalities, and epithelial lesions. Recent advances in molecular research and therapeutic modalities, such as gene and stem cell therapies, show promise for managing diabetic ocular complications. Future research should focus on pathogenetically oriented therapies for diabetic neuropathy and keratopathy, transitioning from animal models to clinical trials to improve patient outcomes.

## 1. Introduction

Diabetes mellitus (DM) is a chronic metabolic disorder characterized by hyperglycemia resulting from defects in insulin secretion, insulin action, or both. With an increasing global prevalence, which has tripled in the last decades [1], DM has emerged as a significant public health concern, causing substantial burdens on individuals, healthcare systems, and economies worldwide. Beyond its well-documented systemic complications affecting multiple organ systems, DM causes important effects on ocular health, with ocular surface disease (OSD) representing a clinically relevant manifestation [2].

OSD encompasses a spectrum of ocular surface disorders characterized by abnormalities in the tear film, cornea, conjunctiva, and meibomian glands, leading to symptoms of ocular discomfort, visual disturbances, and potential vision-threatening complications [3]. While OSD can occur in various clinical contexts, its association with DM has gained increasing recognition in recent years. Mounting evidence suggests that DM contributes to the pathogenesis and exacerbation of OSD through multiple interconnected mechanisms, including alterations in tear film composition, corneal epithelial abnormalities, inflammation, neuropathy, and vascular dysfunction [2,4].

The ocular manifestations of DM, collectively termed diabetic eye disease, present a range of conditions, including diabetic retinopathy, diabetic macular edema, cataract, and secondary glaucoma. While diabetic retinopathy has traditionally received the most attention due to its association with vision loss, the impact of DM on the ocular surface is increasingly appreciated as a significant contributor to visual morbidity and diminished quality of life in individuals with DM [2].

Despite growing recognition of the clinical significance of OSD in DM, several challenges persist in understanding its pathophysiology, diagnosis, and management. Furthermore, the relationship between systemic glycemic control, ocular surface health, and ocular complications of DM remains complex, necessitating a comprehensive approach to each patient [1,2,3]. Advanced glycation end-product accumulation, reduced neurotrophic innervations, dysregulated growth factor signaling, and inflammatory modifications are all part of the pathophysiology of diabetes-related OSD. The aberrant mitochondrial metabolism of the lacrimal gland brought on by the sympathetic nervous system’s hyperactivation may be linked to diabetic dry eye. More research on the immune alterations will identify the predominant pathogenic mechanisms and create targeted intervention strategies for diabetic ocular surface complications, given the significant roles that the dense innervations play in maintaining the homeostatic integrity of the cornea and lacrimal gland [2,4].

In this context, this review aims to provide a comprehensive overview of the epidemiology, pathophysiology, clinical manifestations, diagnostic approaches, and management strategies of diabetes-related ocular surface disease. By discussing the links between DM and OSD and exploring emerging therapeutic modalities, this review seeks to enhance our understanding of this multifaceted clinical entity and optimize patient outcomes in the diabetic population.

## 2. Anatomy and Pathophysiology of the Ocular Surface

The ocular surface comprises the cornea, conjunctiva, and associated adnexal structures, such as the eyelids and tear film. The lacrimal functional unit (LFU) is comprised of the lacrimal apparatus and its associated innervation (motor and sensory fibers) and any changes to one of these elements may have an impact on the others’ proper functioning and homeostasis. LFU is composed of multiple components that function together to control the tear film’s volume and structure by triggering corneal sensory inputs that undergo further processing by the secretory apparatus [5]. This helps to preserve the functionality of the corneal surface. Immune defense processes involve the ocular surface in a complex way, coordinating a fine balance between tolerance to harmless antigens and protection against infections [6].

The conjunctiva, with its specialized immune-associated structures such as lymphoid follicles is assumed to be crucial for the presentation of local antigens and the development of immunological tolerance. Conjunctival-associated lymphoid tissue (CALT) and lacrimal drainage-associated lymphoid tissues (LDALTs) are more structured, organized tissues producing follicles [7,8]. CALT contains a variety of immune cells, including T cells, B cells, dendritic cells, and antigen-presenting cells, which play key roles in initiating and regulating immune responses. In mice, CALT located in the nictating membrane can be overlooked in histology examinations, yet in vivo confocal microscopy (IVCM) has demonstrated that antigen administration induces CALT, with experimental dry eye disease demonstrating a reduction in CALT and allergic conjunctivitis models leading to an increase in CALT [9,10,11]. Conjunctival goblet cells are highly responsive to the cytokine milieu. Research has demonstrated that, analogous to the lung [12], the Th-2 cytokine interleukin (IL)-13 plays a role in the homeostatic regulation of conjunctival goblet cells, while trace amounts of interpheron (IFN)-γ can result in goblet cell death or metaplasia and disrupt mucin secretion [13,14].

The tear film is an elaborate three-layer model fluid, with its components continuously being formed and renewed: lipid layer, mucous layer, and aqueous layer. Conjunctival goblet cells secrete mucins that lubricate and shield the ocular surface. Additionally, they produce immunomodulatory substances including retinoic acid and transforming growth factor-beta (TGF-β), which have been linked to reducing the immunological stimulation of conjunctival antigen-presenting cells [15]. The lipid component of the tear film is produced by meibomian glands, which are holocrine glands, while the aqueous component is produced by the lacrimal gland. Tear film components, including immunoglobulins (IgA and IgG), antimicrobial peptides, and cytokines, contribute to the ocular surface’s innate immune defenses. IgA, predominantly secreted by plasma cells in the lacrimal gland, provides mucosal immunity by neutralizing pathogens and toxins. IgG, derived from serum and local production, offers additional protection against pathogens.

The cornea, traditionally considered immune-privileged due to its avascularity and low immunogenicity, actively participates in immune surveillance and regulation. Corneal epithelial cells and resident immune cells, such as Langerhans cells, dendritic cells, mast cells, macrophages, and lymphocytes, function in antigen presentation and immune modulation [16]. Additionally, the cornea expresses immunomodulatory molecules, such as Fas ligand and programmed death-ligand 1 (PD-L1), which contribute to immune tolerance and the regulation of inflammatory responses. Ocular immune privilege is maintained through various mechanisms, including the blood–ocular barrier, which restricts the entry of immune cells and soluble mediators into ocular tissues, and the expression of immunosuppressive factors, such as TGF-β and IL-10, which inhibit immune activation and inflammation [17]. The identification of innate lymphoid cells (ILCs) represents a major immunological advance in the last few years [18]. These cells lack T- and B-cell antigen-specific receptors and are composed of several heterogeneous lymphoid cell subsets, and they react rapidly to invasive infections and wounds. A recent examination of the ILC population’s embryonic stages revealed that the ILCs were essentially composed of two significant populations: noncytotoxic ILCs and cytotoxic natural killer cells (NKs) [19]. There has been limited research on the presence and properties of ILCs in ocular tissue, despite the fact that similar features have been documented in many other tissues. Numerous recent investigations, including those conducted in our labs, have verified the presence of NK cell subsets in the corneal limbi and normal conjunctiva [16]. A novel theory stated that NK cells might promote corneal repair indirectly by limiting neutrophil recruitment and tissue damage [20]. The dysregulation of ocular surface immunity can lead to the development of inflammatory conditions such as dry eye disease, allergic conjunctivitis, and autoimmune disorders.

## 3. Corneal Alterations and Pathogenesis in Diabetes Mellitus

According to recent research, immune cellular alterations and compromised epithelium wound recovery in diabetic corneas may be signs of a more extensive illness. This emphasizes how vital it is to create novel screening instruments to aid in the early diagnosis and treatment of DM. Any mechanism that influences the rate at which the corneal epithelium regenerates or heals wounds may have important physiological ramifications and adversely affect an individual’s quality of life [21]. DM has been linked in studies to changes in glycosaminoglycan metabolism, aberrant collagen synthesis, reduced oxygen consumption, hypertrophy of the basement membrane, and LFU dysfunction [22] (Figure 1). Neurotrophic ulcer or delayed epithelium healing can be caused by a loss of nerve innervation. Numerous cytokines, chemokines, and cell adhesion molecules are expressed in response to persistent hyperglycemia [23]. A summary of corneal and tear film changes is shown in Table 1.

### 3.1. Corneal Epithelial Alterations

The corneal epithelium is an avascular tissue that serves as a structural barrier to protect the cornea, by blocking the unrestricted movement of fluids from tears and preventing microorganisms from penetrating the corneal epithelium and stroma [24]. In the study by Chang et al. [25], diabetic patients showed alterations in the epithelium, such as a decrease in corneal epithelial basal cell density, sectorial thinning, an increase in cell size variation, bullae formation, and a greater intracellular gap [26]. Furthermore, they revealed a strong correlation between the decrease in epithelial cell density and the densities of nerve branches and nerve fibers. Additional research demonstrated alterations in the epithelium with high glucose levels, which may be helpful in the early identification of corneal epithelium damage [27]. The XYZ hypothesis, where X stands for limbal basal cell proliferation and stratification, Y for basal cell centripetal migration, and Z for superficial cell decomposition, is the most widely accepted explanation for corneal homeostasis. The cornea’s health is contingent upon the total of X and Y equaling Z [28]. In theory, DM causes impairments in the restoration phase of the process, which may result in epithelial thinning in all diseased corneas. However, several investigations have demonstrated that this is not the case, on the contrary [29]. Only in situations of severe diabetes is the epithelium thinner, according to several research studies [30].

In the corneas of streptozotocin (STZ)-induced diabetes in rats, elevated glucose levels disrupted the epidermal growth factor receptor (EGFR) pathway, which eventually resulted in prolonged wound repair [31]. Furthermore, normal epithelial cells respond quickly and immediately to high-glucose treatments that mimic diabetes’ hyperglycemia, which slows down the healing process and reduces cell adhesion [32,33]. During the wound repair phase, corneal epithelial cells exposed to high glucose showed increased matrix metalloproteinase (MMP) activity [34]. MMP is thought to be involved in the alterations seen in diabetic corneas, which include inadequate and delayed epithelial wound repair as well as fractured and fragile basement membranes. Advanced glycation end products (AGEs), 8-hydroxy-2′-deoxyguanosine, and nuclear factor kappa B were shown by Kim et al. to be present in greater amounts in the diabetic rats’ cornea than in the control group [35]. This suggests that these products likely played a role in apoptosis and the subsequent alterations in the cornea that are linked to keratopathy. Several cytokines and growth factors, such as TGF-β, epidermal growth factor (EGF), platelet-derived growth factor (PDGF), thymosin-β4 (Tβ4), IL-6, and IL-10, are important participants in the process of wound healing. These factors affect crucial elements such as cell movement, development, differentiation, survival, and death. The importance of these targets as possible targets for therapeutic interventions in the context of improving epithelial wound healing is highlighted in several studies [26,36].

### 3.2. Corneal Stromal Alterations

The corneal stroma may undergo structural and functional changes as a result of DM, which can impair the patient’s vision by causing a loss of corneal transparency [22]. The diabetic corneal stroma contains abnormal collagen fibrillar bundles that vary in thickness and distribution. Additionally, throughout the entire stroma, AGE immune reactivity may be exhibited, suggesting that this pattern may be caused by collagen crosslinking [37]. An EGFR inhibitor could correct corneal stromal abnormalities by regulating AGE levels. Specifically, it undoes the aberrant behavior of proteoglycans and collagen fibrils. This research implies that the onset of DM-induced corneal stroma remodeling is facilitated by the EGFR signal pathway [38]. Prior studies have identified central corneal thickness (CCT) as a potential biomarker of the diabetic cornea. Anterior Segment Optical Coherence Tomography (AS-OCT), IVCM, or Very High Frequency Ultrasound (VHF-US) have all been used to measure CCT, with the latter being used very infrequently [38]. Several studies have studied CCT in diabetic patients, but the results have been inconsistent. According to D’Andrea et al. [29] and Lee et al. [39], diabetic patients’ CCT was substantially thicker than that of controls. Patients with DM whose glucose levels were within the target therapeutic range and who showed no signs of diabetic retinopathy were included in their study. However, Wiemer et al. [40] and Busted et al. [41] found no correlation between CCT and the degree of retinopathy, duration of the disease, or both. Additionally, Busted et al. found no evidence of a significant relationship between CCT and the length of diabetes, blood glucose levels, or insulin use. A rise in CCT has been linked to the degree of diabetic neuropathy, which is thought to be caused by the rise in stromal thickness [42].

The strengthened adhesive connection between the Descemet membrane and the corneal stroma observed by Schwartz et al. may be connected to high glucose levels, and it may offer a new route for further research [43]. Furthermore, Priyadarsini et al. [44] discovered putative new biomarkers in the corneal stroma (such as aminoadipic acid, pipecolic acid, and dihydroorotate). The stromal response to persistent hyperglycemic stress could involve these prospective biomarkers, as evidenced by their considerable up-regulation in diabetic corneas. These indicators could suggest DM-induced stromal damage, enabling the early detection of complications.

### 3.3. Corneal Endothelial Alterations

Endothelial cell dysfunction and elevated corneal hydration in DM may impact the thickness of the cornea. This may be associated with endothelial issues, which could lead to endothelial decompensation after cataract procedures. Overall, endothelial diabetic alterations appear to be very mild. Numerous investigations have assessed the structure and functionality of the corneal endothelium in individuals with DM. It has been shown that diabetics have an altered endothelial cell shape, exhibiting greater pleomorphism and variation [45]. While more recent research demonstrates lower in vivo cell numbers [46], some data show no modification in density [47]. Endothelial cell function is also the subject of debate; while some research has reported reduced endothelial function, other investigations have not supported this finding [31]. Studying a potential link between the diabetic corneal endothelium and the diabetic neuropathy, dell’Omo et al. discovered no connection between these two complications [48]. The rationale behind these findings is that, while corneal endothelial cell failure is a chronic condition associated with DM, neuropathy is an early manifestation of the disease [49].

Numerous proinflammatory and immunological factors, including MMP, ILs, TNF-α, and vascular endothelial growth factor (VEGF), are found within the endothelial cells. In addition to altering endothelial morphology and function, these stimuli may also damage endothelium and cause molecular changes. Notably, endothelial cell regeneration becomes impaired and slower, and the corneal endothelium barrier’s functionality is compromised [50].

### 3.4. Tear Film Alterations

According to studies, patients with DM have deteriorating tear film function and decreased tear output as their disease progresses [51]. Patients with diabetes have altered tear film in quantity and quality, which increases the risk of dry eye disease (DED), a condition highly prevalent in diabetics [52,53]. Changes in tear film homeostasis have been demonstrated to correlate with the severity of DM and appear to be inversely connected with the duration of the disease [54]. In patients with DM, a research study by Kalaivani et al. [55] revealed a 51.8% prevalence of DED. Studies by Sarkar et al. [56] and Manaviat et al. [57] reported 42% and 54% of cases with DED, respectively. Diabetes-related neuropathy and prolonged inflammation are key factors in DED. The primary pathogenic mechanism behind the abnormalities of tear film is persistent hyperglycemia. Furthermore, diabetic patients’ tears and conjunctiva showed a considerable increase in cytokines such IL-1α, IL-1β, IL-6, IL-8, and TNF-α [22,54]. MMP plays a significant role in tissue damage and low-grade inflammation. Elevated MMP-9 was found to be substantially linked with prolonged ocular surface inflammation [58]. According to a study, diabetic patients may have ocular impairment indicated by an increasing amount of MMP in their tears [59]. Furthermore, in mice, oxidative stress results in pathogenic modification of the acinar cells in the lacrimal gland. Another research revealed that DED was clearly associated with silent mating type information regulation 2 homolog 1 (SIRT1) overexpression [60]. Sirt1 expression has an impact on the endoplasmic reticulum stress pathway’s regulation, slows the degeneration of corneal epithelial cells, and modulates the pathological course of diabetic keratopathy [60].

According to one study, there was a favorable relationship between IgA levels and neutrophil-to-lymphocyte ratio (NLR) values in patients with DED. The same patients also showed a strong association between their C-reactive protein and IgA levels. These findings imply that IgA might be a factor in the low-grade chronic systemic inflammation in DM and that it might be employed as a biomarker for inflammation [61].

### 3.5. Diabetic Corneal Neuropathy

Persistent hyperglycemia triggers a number of degenerative processes affecting peripheral nerves, including the corneal nerves, and causing neuronal degeneration [62]. Diabetes-related nerve damage is caused by a variety of pathological pathways that have been identified. Vascular and metabolic mechanisms interact throughout the entire course of the disease to cause synergistic effects. The polyol pathway, increased production of AGEs, oxidative stress, and protein kinase C (PKC) activation are among the processes and mechanisms that lead to DM consequences shown in Figure 2 [63]. With worsening diabetic neuropathy, a decrease in corneal nerve fiber length, nerve fiber density, and nerve fiber branch density has been observed [64,65].

Edwards et al. and Jiang et al. demonstrated that subjects with diabetic peripheral neuropathy had considerably shorter corneal nerve fibers [66,67]. Misra et al. reported that changes in subbasal nerve density occur before other clinical and electrophysiological investigations of neuropathy [68], and Pritchard et al. demonstrated that a decrease in corneal nerve fiber length is predictive of the onset of neuropathy [69]. Despite the possibility that diabetic neuropathy is linked to decreased nerve migration, recent research has shown that diabetics’ corneal nerves can regenerate following therapies to improve glycemic levels. The length of the nerve fibers significantly improved 12 months afterward. The regulation of limbal epithelial stem cell marker expression with simultaneous regeneration of nerve parameters and significant developments in epithelial regeneration following treatment with various interventions emphasize the importance of corneal innervation in modifying stem cell function in diabetic corneas. Treatment options include genetic [70,71], biological [72], pharmacological [73], and surgical interventions [74]. Major developments in intraepidermal nerve fiber density were observed following two years of rigorous glucose control and lifestyle modifications in patients with type 1 DM who underwent pancreatic and renal transplantation [75].

## 4. Novel Diagnostic Approaches

For a wide range of ocular illnesses, imaging technologies are becoming more and more important and a big component of therapeutic practice. Imaging methods for the ocular surface are not as well established as those for the posterior segment, where OCT has emerged as the gold standard for the diagnosis and monitoring of numerous conditions. Even though the methods discussed show promise, they nevertheless come with higher costs than clinical standard tests performed under a slit lamp. Better documentation and objective assessment are provided by imaging, which is advantageous for long-term management and can also be utilized to educate patients [76].

A fast, non-invasive, and reliable imaging method for measuring small nerve fiber damage is corneal confocal microscopy. In patients with early and established diabetic neuropathy, ocular small nerve fiber loss can be detected using this technique [77]. However, due to a small sample size and differences in image acquisition and processing procedures, several studies have not been able to establish corneal nerve fiber loss in individuals with or without diabetic neuropathy [78]. A recent study group has developed a convolutional network technique that eliminates the requirement for manual nerve annotation by properly enhancing and segmenting corneal nerve fibers in microscope pictures. An autonomous deep-learning structure was used to improve and extract images of the corneal nerve fibers [79].

Epithelial erosions, also known as superficial punctate keratopathy, are often detected in individuals with OSD and can be seen with a high-resolution OCT system. It is possible to measure and visualize larger corneal abrasions, enabling non-invasive wound healing monitoring [80]. As a result, OCT offers a substitute for using dyes in the clinical evaluation of epithelium abnormalities. It has been suggested that another measure for researching OSD is the evaluation of the tear meniscus, as in [81]. Although there have been earlier methods for measuring tear meniscus thickness with a slit lamp, evaluating this parameter by OCT has various benefits, including avoiding reflex tearing from strong light and offering good repeatability. Meibomian gland imaging has also been performed with OCT, utilizing both commercially available and custom-built equipment [80,81].

Ocular thermography has also been used to distinguish DED from multiple etiologies. Patients with aqueous deficiency dry eye disease appear to exhibit lower ocular surface temperatures but greater interblink loss of energy. In contrast, patients with evaporative dry eye disease appear to exhibit lower cooling rates [82]. The tear film imager is capable of measuring tear physiology immediately and dynamically in a single measurement. It gives images of the cornea with a vast field of view and a high lateral resolution in the nanoscale range utilizing spectral interference technology, which allows for normal eye activity, such as blinks, saccadic motions, and focusing/defocusing events. The tear film imager makes it possible to measure the thickness of the lipid and mucous-aqueous layer [83]. Lipid layer thickness can be measured on a very small scale using ocular surface interferometry. A white light-emitting diode light source illuminates the ocular surface, and a camera records the resulting image. Then, it is possible to link particular lipid layer thickness values with optical interference colors on the tear film’s surface [84]. This is a significant improvement over previously employed methods for lipid layer evaluation that were likewise based on interferometry but only offered qualitative grading with a slit lamp [85].

## 5. Updates on Treatment Options

Pharmacological medication for diabetes should be the initial treatment for corneal neuropathy in order to achieve precise blood glucose control. Systemic therapy is the most important tool for treating DM and preventing its complications when lifestyle and diet changes are not effective [86]. Effective management must limit further advancement of corneal neuropathy. In order to effectively treat diabetic keratopathy, epithelial repair must be accomplished. Unfortunately, there are currently no pathogenetically oriented pharmaceutical treatments available for neurotrophic keratopathy.

Novel systemic medications have been introduced (Table 2), among which is Teplizumab, a humanized monoclonal antibody to CD3 on T cells [87], a long-lasting GLP-1/anti-GLP-1R antibody fusion protein (Glutazumab) [88], SGLt-2 inhibitors (Canagliflozin and Dapagliflozin), Technosphere insulin [89], beta-carotene, which has been shown to ameliorate the ultrastructural alterations in the cornea associated with DM [90], oral treatment with esolvin-D1, an anti-inflammatory eicosanoid, along with fish oil (which has been shown to reduce degeneration of nerves), and a combination of menhaden oil, α-lipoic acid, and enalapril, which has been shown to reverse diabetic corneal and peripheral neuropathy [91].

Another method for treating diabetic keratopathy is gene therapy. This approach aims to identify a single, highly specific disease-associated target and modify it by either increasing or decreasing the activity of the abnormal designated gene [99] (Table 3). It was reported that in human organ-cultured diabetic corneas, adenoviral gene therapy addressed epithelium and stem cell marker expression by overexpressing the c-met proto-oncogene and/or suppressing MMP-10 and cathepsin F [91]. This therapy sped up wound healing even when it solely transduced the limbal stem cell compartment.

Another study detected epigenetic differences between diabetic and non-diabetic human limbal epithelial cells abundant in stem cells, as well as other diabetic indicators that can be addressed to regulate corneal epithelial wound regeneration and stem cell proliferation [70]. There were substantial differences in DNA methylation among the two study groups. The WNT5A promoter was hypermethylated in diabetic limbal epithelial cells, followed by a significant decrease in Wnt-5a protein. Exogenous Wnt-5a treatment of diabetic corneas improved the healing process while also increasing the levels of limbal epithelial stem cells [70]. The miR-203a antagomir inhibitor improved Wnt5a expression and accelerated the healing process in diabetic LEC. They developed a nanoconjugate containing antisense oligonucleotides and miR-203a to stimulate Wnt5a activity and tested its capacity to improve wound healing in diabetic corneal epithelial cells [71]. These findings emphasize the functional importance of the limbal cells niche segment in healing corneal abnormalities in diabetic patients.

Novel molecular approaches are currently being utilized to identify the limbal epithelium stem cell phenotype in DM. Although most research demonstrates changes in limbal cell biomarker activity, transcriptional investigations employing single-cell ribonucleic acid (RNA) sequencing showed little difference between diabetic and normal rat corneas [104]. Particularly, no differences in mRNA transcripts for the limbal cell-associated genes Krt14 and Gpha2 were found in STZ-induced type I DM compared to the control group [104]. In this model, cell heterogeneity was well conserved, with just two genes showing differential expression in DM versus normal cells. Furthermore, it is plausible that limbal epithelial stem cell dysfunction caused by high glucose levels does not appear until the cornea is affected. It is also possible that poor mRNA translation is responsible for low limbal stem cell marker activity in diabetic corneas [105,106]. Differences between research may be due to different animal models, the duration and impact of hyperglycemia, and various biomarkers and investigations used [107].

Stem cells treatments for corneal pathology are under investigation, with promising outcomes. In rats, the injection of mesenchymal and hematopoietic stem cells stimulates corneal epithelial restoration [42]. Furthermore, compared to the control group, transplanting stem cells in rat models with induced diabetic keratopathy enhances corneal reepithelialization with fewer vacuolated cells. Pathophysiologically, these improving effects are linked to increased production of TSG-6 (tumor necrosis factor-α-stimulated gene/protein-6) in the corneal epithelium, which controls cell motility and extracellular matrix stability [106,107].

For patients with diabetic neuropathy, there are comparatively limited therapeutic options available to control neuronal functioning and restore corneal innervations. However, some research has suggested that the application of neurotrophic factors could be a viable treatment for the regeneration of the corneal nerves. Through their receptors in the epithelial layer, factors like nerve growth factor (NGF), glial cell-derived neurotrophic factor, and ciliary neurotrophic factor have been shown to stimulate corneal epithelial cell growth and migration as well as to improve their immunoprotective activity [91,108]. In both in vitro and in vivo diabetic models, the local application of NGF promoted the generation of reactive oxygen species (ROS), which in turn reversed the apoptosis and inflammation in the cornea [94]. Various growth factors and axon guidance molecules, such as Semaphorins, are upregulated during cornea injury. One study observed that DM-induced mice subjected to corneal epithelial debridement and receiving intrastromal micropocket implantation with Semaphorin showed improved corneal nerve regeneration compared to the control group [93].

Topical corticosteroids are successful at breaking the cycle of inflammation because they are strong inhibitors of several inflammatory mediators. Acute-phase cytokines, including IL-1 and TNF-α, ICAM-1, MMPs, chemokines, prostaglandins, and phospholipase A, are all suppressed when NF-κB is suppressed. Moreover, local corticosteroids lessen the influx of leukocytes into inflammatory ocular tissues [109]. The ophthalmic corticosteroid loteprednol etabonate was designed to quickly degrade after being administered, lowering the potential risks associated with other therapies. KPI-121 is an ocular suspension that more effectively delivers loteprednol etabonate to the ocular tissues by utilizing MPP technology. With its innovative drug delivery mechanism, the recently approved KPI-121 0.25% shows promise in managing the recurrent episodes of dry eye that many diabetic patients encounter [110].

It has been demonstrated that autologous blood serum topical treatment is beneficial for DED. The important elements found in tears are immunoglobulins, vitamin A, fibronectin, growth factors, and anti-inflammatory cytokines. Retrospective cohort research has shown that 50% of eye drops are safe and effective for treating severe dry eye that is unresponsive to all available traditional therapies [111]. Investigations are being conducted on other anti-inflammation treatments for the treatment of DED. Tβ4, an essential component of thrombocytes, macrophages, and polymorphonuclear cells with multiple regenerative properties and involved in wound healing and trauma reaction, is an amino acid protein that is synthesized and present in RGN-259 ocular solution. In a rat model with DED, Tβ4 was observed to speed up reepithelialization, diminish inflammation, and enhance stem cell migration [92]. Additionally, prompt intervention to corneal injuries and the avoidance of mechanical and environmental risk factors that might predispose patients corneal infections contribute to preventive efforts [112].

For neurotrophic keratitis, there is one FDA-approved medication, oxydervate (cenegermin). When compared to a placebo group, recombinant human nerve growth factor oxervate was found to ameliorate keratitis in 70% of participants. Aldose reductase inhibitors (ARIs), opioid growth factors, hepatocyte growth factors (HGFs), and other local treatments for neurotrophic keratitis are also being investigated in current research [113]. HGF-mediated immunosuppression and accelerated epithelial cell multiplication during corneal wound repair may be attributed to higher HGF signaling, as suggested by increased HGF-R expression in HGF-treated corneas. Studies have demonstrated that HGF inhibits the expression of α-smooth muscle actin in corneal fibroblasts, which is triggered by TGF-β, hence preventing fibrosis in patients with epithelial damage [114]. However, palliative care—which relies heavily on symptoms and medications mostly studied in individuals without DM—remains the standard of care for diabetic patients with ocular surface disease [115].

## 6. Future Perspectives

In the last few years, DM has become a more widely recognized disease that impacts every structure in the eye. DM can present with a variety of clinical symptoms, the most common of which are neuropathy, abnormalities of the tear film, and epithelial lesions. The underlying molecular processes of DK are still unknown. This paper provides an overview of the various underlying pathophysiologic mechanisms involved in corneal alterations. A number of innovative and precise techniques have been developed to study changes in the diabetic cornea. To manage DM as best as possible, a better understanding of the etiology and changes associated with DM as well as the investigation of possible biomarkers would be beneficial. Targeted molecular therapies (such as gene and stem cell therapies) and other recently reported therapeutic agents seem promising, even if there is still no consistent standard of care accessible for treatment of the cornea in diabetic patients. One of the key goals of future research should be to build pathogenetically oriented therapy for diabetic neuropathy and keratopathy by transitioning from animal models to clinical trials.

The most important research gap in current developments of DM management is translation from animal to human clinical trials. Several animal models allow for a more accurate representation of the relatively early symptoms of diabetic corneal problems in humans, such as diabetic keratopathy and neuropathy. However, the primary limitation of present research in preclinical trials of novel medications is the predominance of type 1 DM induced animal models, despite the fact that type 2 DM affects the majority of people with the disease. Furthermore, it has been demonstrated that certain characteristics of diabetes in the cornea of animals differ from the condition in humans [116]. Although the features of diabetic keratopathy in humans and animals differ, the use of animal models has aided in our understanding of the condition and in the search for improved treatments. Because the existing in vivo models closely resemble human corneas, the data can be efficiently translated into clinical trials.

## 7. Conclusions

Ocular surface disease in DM is a multifactorial disease that is described by an important and sustained inflammatory response that can lead to chronic complications. Progress in discerning the underlying immunopathological changes within the corneal layers and tear film has led to major advances in local and systemic therapy that target specific inflammatory effectors and pathways. These treatments are of utmost importance in order to prevent chronic disease and progression.

## Figures and Tables

**Figure 1 biomedicines-12-01303-f001:**
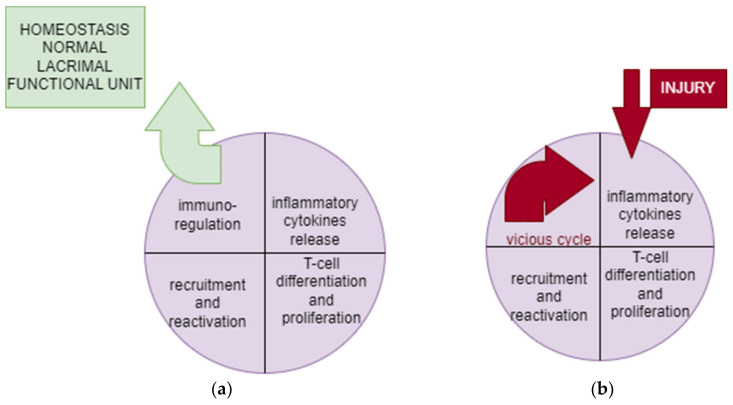
(**a**) Normal adaptive immune ocular response. (**b**) Dysregulated adaptive immune ocular response.

**Figure 2 biomedicines-12-01303-f002:**
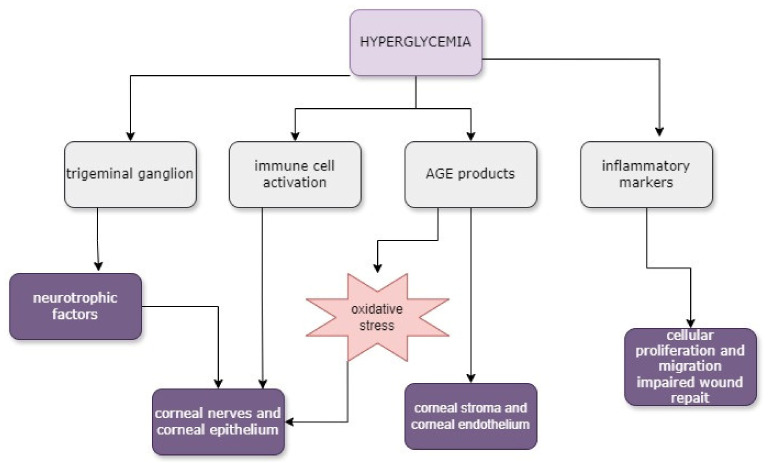
Mechanisms of persistent hyperglycemia leading to degenerative processes of ocular surface. Abbreviation: AGE-advanced glycation end-products.

**Table 1 biomedicines-12-01303-t001:** Diabetic structural damage on corneal layers and tear film.

Ocular Structure	Diabetic Alteration
Epithelium	Increased susceptibility to damage, delayed wound healing, epithelial defects
Stroma	Corneal edema, decreased sensitivity, alterations in collagen structure
Endothelium	Endothelial dysfunction, decreased cell density, increased risk of corneal edema
Tear film	Reduced tear secretion, altered composition (e.g., increased osmolarity)

**Table 2 biomedicines-12-01303-t002:** Advances in the treatment of diabetes mellitus and ocular surface disease.

Treatment	Route of Administration	Current State of Development
Teplizumab [87]	intravenous infusion	FDA approved (2022)
Glutazumab [88]	intravenous injection	clinical trials, approved in a few countries
Canaglifolzin [91]	oral	FDA approved (2013)
Dapaglifolzin [91]	oral	FDA approved (2014)
Technosphere insulin [89]	subcutaneous bolus	FDA approved (2014)
Beta-carotene [90]	systemic	human clinical trials
Thymosin beta 4 [92]	eye drops	phase III clinical trial
Semaphorins [93]	intrastromal injection	clinical trials
Nerve growth factor [94]	eye drops	clinical trials
Epalerestat [95]	silicon hydrogel contact lens	animal trials
Substance P [96]	eye drops	animal trials
Ciliary neurotrophic factor [97]	subconjunctival injection	animal trials
Fibronectin-derived peptide PHSRN [98]	eye drops	animal trials

Abbreviations: FDA—Food and Drug Administration; PHSRN—Pro–His–Ser–Arg–Asn.

**Table 3 biomedicines-12-01303-t003:** Gene therapy studies.

Study	Genetic/Epigenetic Target	Mechanisms
Leszczynska et al. [100]	limbal stromal cell-derived exosomes	When primary limbal epithelial cells are cultured with exosomes, their proliferation rate is noticeably higher than that of untreated cells. Wound healing is importantly increased.
Shah et al. [70]	Wnt-5a	Improved the healing process while also increasing the levels of limbal epithelial stem cells.
Hu et al. [101]	miR-34c	By directly interacting with antigen4B, miR-34c influences both the development of trigeminal sensory neurons and the healing of diabetic corneal nerve endings. Subconjunctival injection may promote corneal epithelial healing.
Kramerov [71]	miR-203a antagomir inhibitor	Improved Wnt5a expression and accelerated the healing process in limbal epithelial cells.
Bikbova [91]	recombinant adenovirus (rAV)-driven small hairpin RNA (rAV-sh)	Improved wound healing, corneal epithelial and limbal stem cells. Even more effective when in a combined treatment with the overexpression of C-met.
Kulkarni et al. [102]	miR-10b	Regulated limbal epithelial stem cells’ early proliferation state during renewal and division, improved stem cell function and corneal homeostasis.
Herencia-Bueno et al. [103]	histone H3	Reduced chromatin compaction changes in both epithelial and stromal cells, as well as a decrease in histone H3 acetylation, were seen in diabetic rats’ corneal sensitivity.

## Data Availability

The data presented in this study are available on request from the corresponding author.

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
