# Peer review of "Immune-Mediated Ocular Surface Disease in Diabetes Mellitus—Clinical Perspectives and Treatment: A Narrative Review"

_biomedicines, 2024, doi:10.3390/biomedicines12061303_

Round 1

Reviewer 1 Report

Comments and Suggestions for Authors

The manuscript presents a comprehensive review of ocular surface disease (OSD) in the context of diabetes mellitus (DM), highlighting its epidemiology, pathophysiology, clinical manifestations, diagnostic approaches, and management strategies. However, there are several areas where the review could be strengthened:

  1. Current Research Gaps: The discussion on recent advances in molecular research and therapeutic modalities, such as gene and stem cell therapies, is promising. Expanding on the current research gaps, particularly in the translation from animal models to human clinical trials, would enrich the review.

  2. Diagnostic Challenges: While the review mentions diagnostic approaches to diabetes-related OSD, it lacks detail on the specific challenges and limitations of current diagnostic techniques. A more detailed discussion on how these challenges can be overcome with newer technologies or methodologies would be beneficial.

Author Response

Dear reviewer,

I appreciate greatly the clear and valuable observations and made all the changes that you suggested. I will address all of them separately in the next reply

  1. We have added a paragraph regarding difficulties in managing the transition between animal to human trials and disadvantages.
  2. We have also added an entirely new chapter regarding only diagnostic options, taking into account novel options, with research articles published in the last few years
  3. We have also added a table with an emphasis on gene therapy and fixed all the typos throughout the manuscript

Once again, thank you very much for this kind review. 

With utmost respect,

Assist.prof. Ghenciu Laura Andreea

Reviewer 2 Report

Comments and Suggestions for Authors

The authors did a very comprehensive review about the immune-mediated ocular surface disease in diabetes mellitus. Several comments from the reviewer are given below to improve the quality of this paper.

1. The quality of Figure 1 is poor, revision of the figure will be needed.

2. In Table 2 , "Abbreviations: 313 FDA-Food and Drug Administration; PHSRN- Pro–His–Ser–Arg–Asn." should be list as a footnote and placed under the atble.

3. In page 8, a table to list the genetic and epigenetic treatment  methods will be more helpful. 

Comments on the Quality of English Language

Minor editing of English language required

Author Response

Dear reviewer,

First of all, thank you for your precious time. We are glad that you find this topic interesting. I will address the problems in the next paragraphs.

  1. We revised Figure 1 and also Figure 2, thank you very much and have also footnoted the abbreviations
  2. We have added a paragraph and a comprehensive table regarding targets using gene therapy. We have also added an entire chapter on novel diagnostic methods in ocular surface disease

Once again, we thank you for your time.

With utmost respect,

Assist.prof. Ghenciu Laura Andreea

Reviewer 3 Report

Comments and Suggestions for Authors

Dear authors, 

The manuscript is well written, and I truly appreciate the opportunity to review the manuscript. Minor revisions are needed for getting the manuscript into a publication form. My comments are attached below. 

I appreciate the opportunity to review this interesting review on “Immune-Mediated Ocular Surface Disease in Diabetes Mellitus”. While the paper addresses an interesting issue, it needs some minor corrections. The introduction written by the authors can be more elaborated. Though the language and sentence structures of this manuscript is well represented but some minor language editing is needed to allow for a proper peer review. Corrections for English language and grammatical errors is required throughout the manuscript. References are missing from many places.

My comments are below:

Figure 1: 1b not clear and needs redrawing

Line 176: MMP is abbreviated but

Line 252: MMP has full form. Please check this

Figure 2: Check Hyperglycemia spelling

Also, the oxidative stress is below the star shape sign. Check this.

Line 320: MMP check

Line 322-327: References missing

Line 335-339: References missing

Line 356-359: References missing

Some places DM used and some place diabetes mellitus full form used. Check this.

Comments on the Quality of English Language

Some corrections of English language required and check for the grammatical errors with thorough language editing. 

Author Response

Dear reviewer,
First of all, thank you for your precious time. We are glad that you find this topic interesting. I will address the problems in the next paragraphs.

  1. We have added context to the introduction, describing the exact pathophysiological changes leading to OSD
  2. we have checked all the typos and references throughout the manuscript, thank you very much
  3. we have also reviewed Figure 1 and 2
  4. we have added a table discussing gene therapy targets, and mechanisms and an entire chapter on novel diagnostic options

Once again, we thank you for your time.
With utmost respect,
Assist.prof. Ghenciu Laura Andreea